# Optical bench simulation for intraocular lenses using field-tracing technology

**Seok Ho Song**[1☯]**, In Seok Song**[2☯]**, Se Jin Oh**[1]**, Hyeck-Soo Son**[3]**, Min Ho Kang**[4,5]*

**1** Department of Physics, Hanyang University College of Natural Science, Seoul, Republic of Korea, **2** Seoul Eye Clinic, Goyang, Republic of Korea, **3** University Eye Clinic of Heidelberg, Heidelberg, Germany, **4** Department of Ophthalmology, Hanyang University Guri Hospital, Guri, Republic of Korea, **5** Department of Ophthalmology, Hanyang University College of Medicine, Seoul, Republic of Korea

☯ These authors contributed equally to this work.
* bsdoc@hanyang.ac.kr, ocularimmunity@gmail.com

## Abstract

### Purpose

To evaluate the image quality of intraocular lenses (IOLs) using field-tracing optical simulation and then compare it with the image quality using conventional ray-tracing simulation.

### Methods

We simulated aspheric IOLs with a decenter, tilt, and no misalignment using an aspheric corneal eye model with a positive spherical aberration. The retinal image, Strehl ratio, and modulation transfer function (MTF) were compared between the ray-tracing and field-tracing optical simulation and confirmed by the results reported in an *in vitro* experiment using the same eye model.

### Results

The retinal image showed interference fringes from target due to diffraction from the object in a field-tracing simulation. When compared with the experimental results, the field tracing represented the experimental results more precisely than ray tracing after passing over 400 μm of the decentration and 4 degrees of the tilt of the IOLs. The MTF values showed similar results for the case of no IOL misalignment in both the field tracing and ray tracing. In the case of the 200-μm decentration or 8-degree tilt of IOL, the field-traced MTF shows lower values than the ray-traced one.

### Conclusions

The field-tracing optical bench simulation is a reliable method to evaluate IOL performance according to the IOL misalignment. It can provide retinal image quality close to real by taking into account the wave nature of light, interference and diffraction to explain to patients having the IOL misalignment.

**Data Availability Statement:** All relevant data are within the manuscript and its Supporting Information files.

**Funding:** The authors have no financial or proprietary interest in the materials presented herein.

**Competing interests:** NO authors have competing interests

## Introduction

Light is regarded as a ray in geometrical optics but an electromagnetic wave in wave optics [1]. Ray tracing assumes that the wavelength of light is sufficiently small so that light propagation can be described in terms of rays. It is fast but not accurate enough for simulation of most micro- and nanostructured components [2]. In field tracing, electromagnetic harmonic fields are traced through the optical system. Field tracing formulates not only the generalization of ray tracing but also of electromagnetic wave modeling [3]. In addition, field tracing of propagating electromagnetic waves allows for a much more accurate description of many optical effects and becomes necessary for small diffractive structures, which cannot be properly modeled by ray tracing [4].

Numerous designs of intraocular lenses (IOLs) for specific purposes, such as spherical, aspheric, toric, or presbyopia-correcting, have been developed. Optical simulation software like CODE V (Optical Research Associates, Pasadena, CA), OSLO (Lambda Research Corporation, Littleton, MA), or ZEMAX (ZEMAX LLC, Kirkland, WA, USA) has been used to investigate the optical quality of these IOLs, design various conditions in an optical environment, and compare results with those of an *in vitro* optical bench test [5–12]. However, all these simulation software presented are based on ray-tracing technology.

By using the wave nature of light, current field-tracing technology enables optical modeling and design [2]. Wave optical simulation software, VirtualLab (Wyrowski Photonics GmbH, Jena, Germany), was recently developed and introduced in many areas of optics [2,13–16]. This field-tracing approach provides simulation techniques covering everything from geometrical optics to electromagnetic field methods in a single platform. An essential part of this simulation technique is the propagation of harmonic fields through homogeneous media [13]. In our current study, we evaluated the image quality of IOLs using field-tracing optical simulation and compared it with the image quality obtained by conventional ray tracing qualitatively and quantitatively according to the IOL misalignment.

## Materials and methods

### Study design

To evaluate the image quality of IOLs, a commercial field tracing program, VirtualLab (Wyrowski Photonics GmbH, Jena, Germany), was used in this work. The field-traced behaviors were compared with those obtained by the well-known ray tracing software from ZEMAX (ZEMAX LLC, Kirkland, WA, USA). IOLs with misalignment errors of decenter and tilt were simulated by these two types of optical simulations. The results were also compared with *in vitro* measurement reported in an experiment by Pieh et al. [17].

**Eye modeling and IOL simulation setup.** A schematic illustration of the optical simulation setup is shown in Fig 1. The setup consists of an artificial cornea and a wet cell containing the IOL in water between two parallel glass plates of optical quality (BK7; Schott, Southbridge, MA). A collimated laser beam (543.5 nm in wavelength) is incident to an aperture stop after diffraction from an object mask (U.S. Air Force resolution target). The aperture stop has a diameter of 9.072 mm for producing a 5-mm pupil on the IOL. The optical parameters of the artificial cornea with an intermediate spherical aberration (SA) and the IOL (SA-correcting, 20 D) were the same as the experimental ones reported in the reference experiment:[17] the intermediate SA cornea was defined by a K value of 43 D, asphericity of -0.26, and SA Z(4,0) of 0.172 μm for a 6-mm aperture, the SA-correcting IOL means a IOL compensating the corneal SA of 0.172 μm. The material of the IOL in the setup is acrylate with a refractive index of 1.47 at room temperature. The IOL can be shifted vertically along the y-axis or rotated about the y-axis to produce decentered or tilted images on the screen.

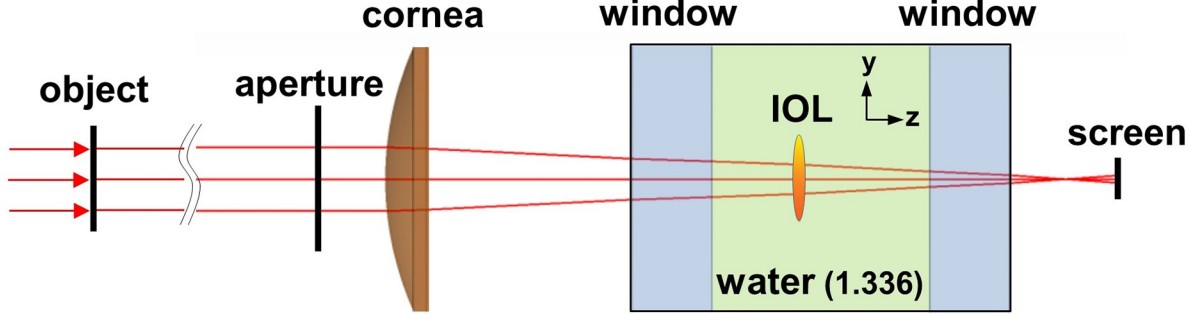

**Fig 1. Diagram of the optical simulation setup.** IOL = intraocular lens.

It is noted that in our field-tracing optical bench simulation we use a rigorous field propagation method, spectrum of plane wave (SPW) between the components in Fig 1. The SPW operator provided in VirtualLab is an angular spectrum integral method that can provide retinal image quality close to real by taking into account the wave nature of light, interference and diffraction.

**IOL decentration and tilt.**   For evaluation of the IOL decentration, the IOL was shifted to a distance of 800 μm from the z-axis in 100 μm steps. For tilt evaluation, the IOL was rotated to an angle of 8 degrees about the y-axis in 2 degree steps. The results of the IOL decentration and tilt simulation were compared with the extraction values from the experimental curves depicted in the reference experiment [17].

**Optical image quality assessment.**   Qualitative and quantitative comparisons of the image and Strehl ratio between the two simulation methods were evaluated and confirmed by the results reported in the reference experiment [17]. The Strehl ratios as a function of the IOL decentered distance and tilted angle were evaluated as a comparative metric for image quality. Before ratio calculation, the screen position must be adjusted to a focal plane after removing the object mask in order to obtain the exact focal spots.

The most widely used measure of the resolving power and image definition produced by the IOLs is the modulation transfer function (MTF). An evaluation using MTF may have better repeatability and reproducibility than previous imaging tests with the resolution target and of the Strehl-ratio test. Therefore, here we also compared the MTF values of the IOL by field tracing and ray tracing, based on the intensity distributions of the point spread functions (PSFs) using a Fourier transformation.

## Results

Fig 2 shows the images of the field-tracing simulation. The 5 mm x 5 mm object in Fig 2(A) is imaged on the screen when the decentered distances of the IOL from the z-axis are 0 μm, 400 μm, and 800 μm, as shown in Fig 2(B)–2(D), respectively. The magnified images (dashed squares) with dimensions of about 200 μm x 200 μm clearly show the interference fringes inside the white bars by diffraction from the object. The blurring effect of the edge boundaries is more severe due to the large decentration in Fig 2(D).

As a comparative metric for image quality, the Strehl ratios as a function of the IOL decentered distance are compared in Fig 2(E), obtained by field-tracing (filled red circle), ray-tracing (open blue circle), and extraction values (square box) from the experimental curves depicted in the reference experiment [17]. The fitting curves of the field-traced and ray-traced ratios show the guidance of their evolutions. In general, the Strehl ratios of all the cases decrease with increasing decentration. The focal spots are gradually crushed toward the shifted direction as

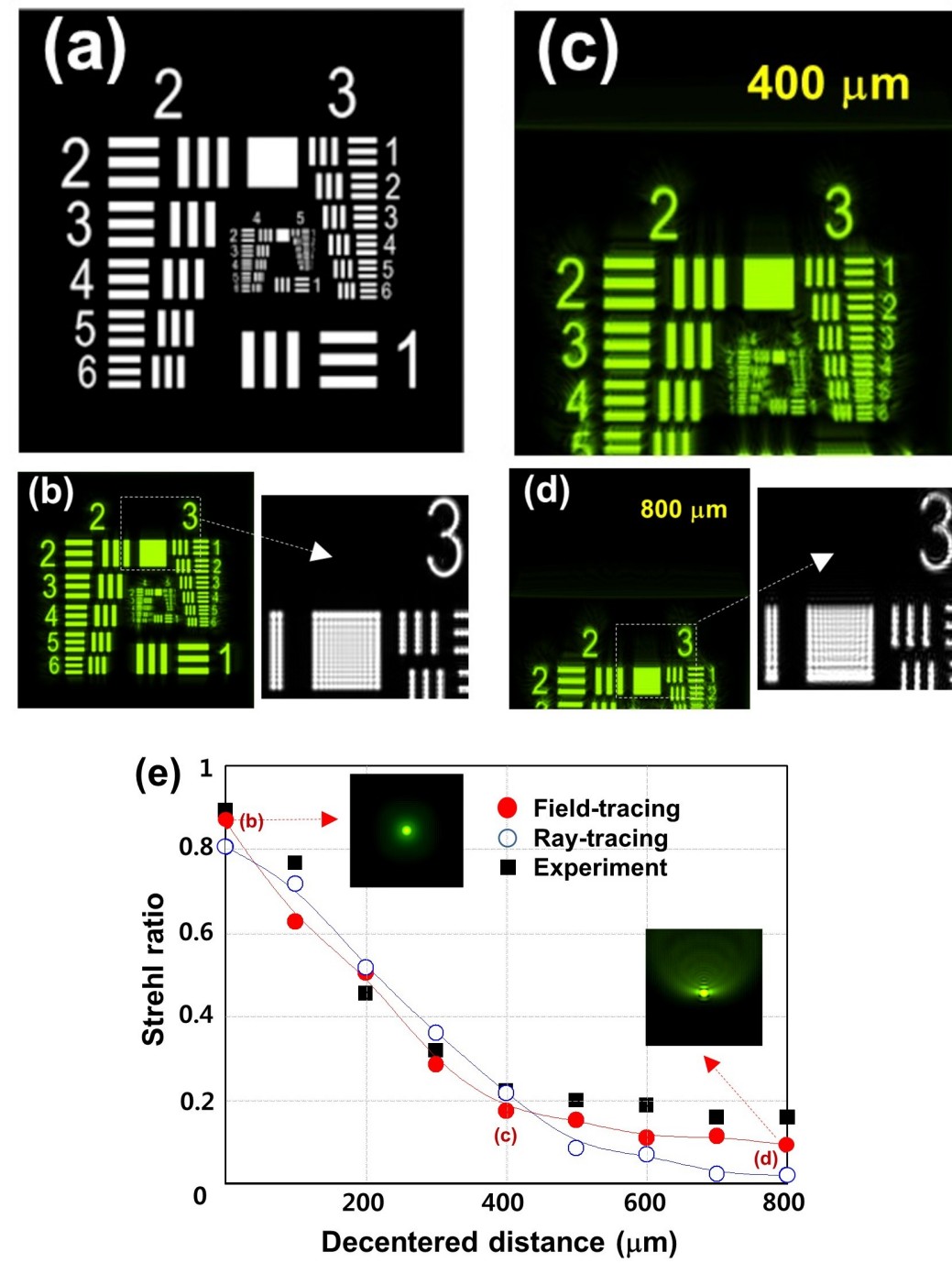

**Fig 2. IOL decentration.** The 5 mm x 5 mm object (**a**) is imaged on the screen using field-tracing simulation when the decentered distances of the IOL from the z-axis are 0 μm (**b**), 400 μm (**c**), and 800 μm (**d**). The magnified images (dashed squares) with dimension of about 200 μm x 200 μm clearly show the interference fringes inside the white bars. (**e**) Strehl ratios obtained by field-tracing (filled red circle), ray-tracing (open blue circle), and extraction values (square box) from the reference experiment. The 2 insets of 2-dimensinal PSFs for 0 μm and 800 μm decentered distances were obtained by field tracing. The 3 positions marked with (**b**)-(**d**) on the field-tracing curve correspond to the images of (**b**)-(**d**).

shown, for example, in the two insets of the 2-dimensional PSFs for 0 μm and 800 μm decentered distances obtained by field tracing. The three positions marked with (b)-(d) on the field-tracing curve correspond to the images in Fig 2(B)–2(D). It is noted that the field-tracing results more closely track down the experimental ones, in particular after 400 μm decentration (below a ratio of 0.2).

For the tilted IOLs, the images of the test target obtained by field-tracing are presented in Fig 3. The tilt angles about the y-axis are 2, 4, 6, and 8 degrees in Fig 3(A)–3(D), respectively, and their Strehl ratios are depicted in Fig 3(E). The right-hand sides closer to the tilted IOL surfaces of the images are more severely smeared as obviously shown at the magnified part in Fig 3(D). It is also noted that the Strehl ratios in Fig 3(E) decrease with increasing tilt angle, and the focal spots are gradually dispersed toward one direction as shown, for example, in the inset of PSF for the 8-degree tilt obtained by field tracing. Again, the field-traced ratios more precisely represent the experimental ones, in particular after a 4-degree tilt (below the ratio of 0.4), as expected.

The MTF values of the IOL by field tracing (solid red curve) and ray tracing (dashed blue curve) are compared in Fig 4. The data is based on the intensity distributions of the PSFs in Figs 2(E) and 3(E) by Fourier transformation. As representative examples, cases of IOLs with no misalignment, 200-μm decentration, and 8-degree tilt should be noted. In the case of no misalignment, similar results were obtained for both field-traced and ray-traced MTF curves. In the case of 200-μm decentration, the field-traced MTF curve shows lower values than the ray-traced one. In the case of 8-degree tilt, the field-traced MTF curve usually shows lower values than the ray-traced one but more rigorous patterns.

## Discussion

In the current study, we evaluated the optical quality of IOLs using ray- and field- tracing optical simulations in the setting of aspheric IOLs with decenter and tilt errors using the aspheric corneal eye model with a positive SA. One can clearly see the interference fringes inside the magnified white bars by diffraction from the object in a field-tracing simulation. This is definite evidence that the field-tracing simulation used in this study can account for the diffraction effect of light. The greater decentration and tilt of the IOLs induce more blurring effects on the images and less Strehl ratios, as has been noted elsewhere [17–19]. When compared with the experimental results, the field-tracing results more precisely represent the experimental ones than the ray-tracing results after decentration of 400 μm and a tilt of 4 degrees. This is, as expected, due to the fact that field-tracing can properly reproduce a retinal image quality close to real, even with severe interference fringes.

The prominent interference fringes in the field-tracing images could be attributed to three reasons. First, the light source presented in the current study was a monochromatic laser, which is very coherent to cause an interference effect of light. Second, the image resolution in the image plane of this study was sufficiently high enough to express a subtle change in the image. Third, field tracing formulates a generalization of the electromagnetic wave modeling, which enables this method to express the diffractive properties of light. The higher requirements concerning the CPU and memory were disadvantages of field tracing in the past, but due to the availability of more powerful computers, these methods have attracted increasing interest in recent years [2,4,13–16].

A comparison of the field-traced ratios with those from the experiment shown in Figs 2 and 3 reveal no perfect accordance. The differences between them may mainly be caused by the inaccuracy of the IOL positions in an experimental setup as mentioned in the work by Dragostinoff et al. [4] Although no perfect accordance was observed, our results clearly show the

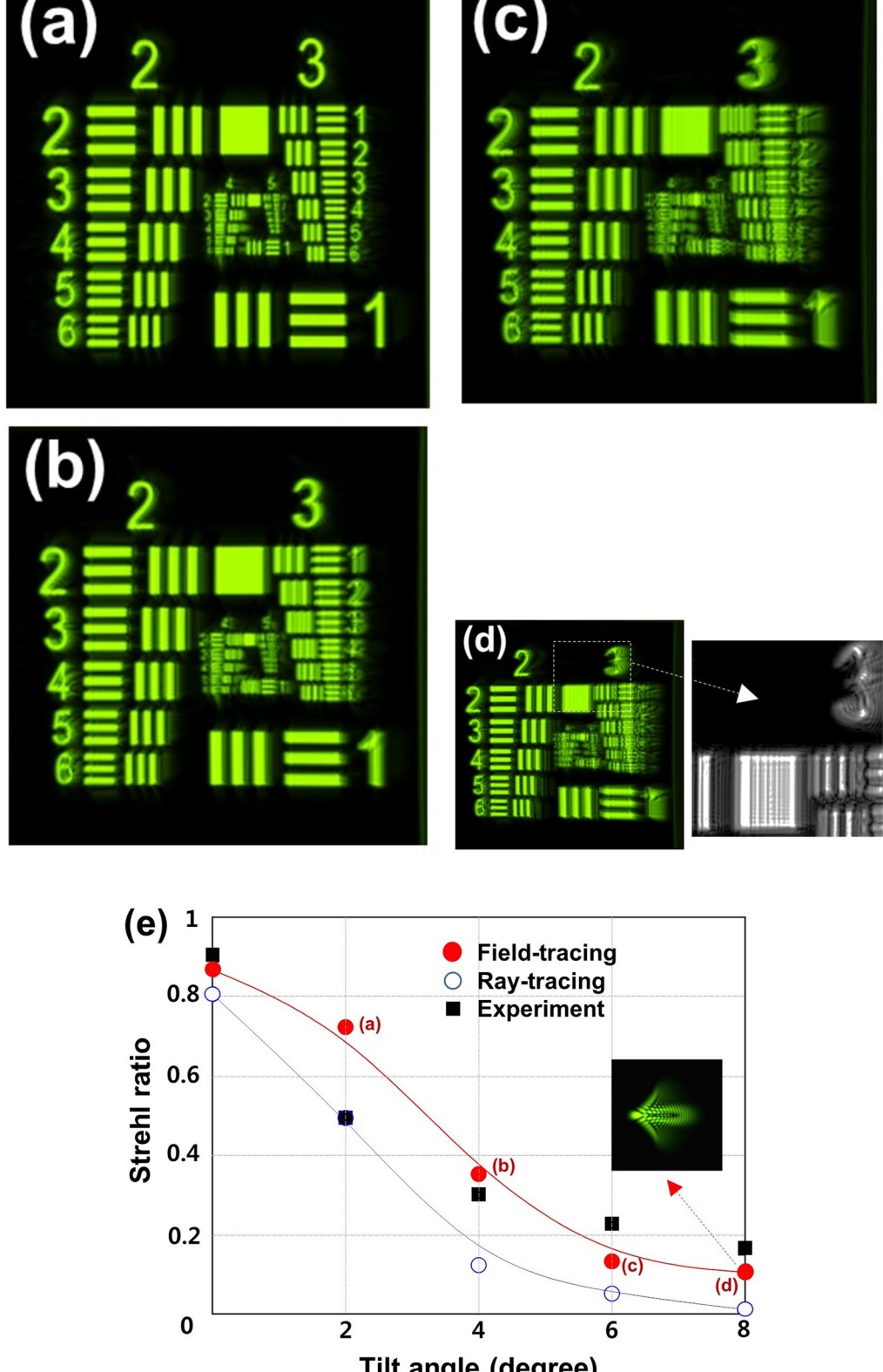

**Fig 3. IOL tilt.** The images of filed tracing simulation with the tilt angles about the y-axis are 2, 4, 6, and 8 degrees (**a,b,c, d**). The magnified image (dashed squares) shows the right-hand sides, closer to the tilted IOL surfaces, of the images are

more severely smeared. (**e**) Strehl ratios obtained by field-tracing (filled red circle), ray-tracing (open blue circle), and extraction values (square box) from the reference experiment. The inset of PSF for 8-degree tilt was obtained by field tracing. The 3 positions marked with (**a**)-(**d**) on the field-tracing curve correspond to the images of (**a**)-(**d**).

superiority of field-tracing techniques. Dragostinoff et al. reported that the results of qualitative behavior and simulated MTF values for a diffractive multifocal IOL using a field-tracing technique demonstrated the sufficient modeling capability of this technique [4].

The MTF values based on the intensity distributions of the PSFs were calculated by Fourier transformation for the case of no misalignment and a 200-μm decentration 8-degree tilt in Fig 4. Similar results between field-tracing and ray-tracing simulations obtained for the case of no misalignment mean that if the aspheric IOL is placed under perfect axial alignment, the diffractive effects are minimal, and the differences between field-traced and ray-traced simulation are negligible. This finding is similar with the results of Dragostinoff et al. However, if the aspheric IOL is placed with misalignment such as decentration or tilt, the diffractive effects are not negligible, so the field-tracing technique degrades the optical quality of IOL more than the ray-tracing technique.

Optical bench tests for IOLs are complementary to clinical assessment because, in addition to being objective and patient independent, they have the ability to control factors that are difficult to address in clinical work such as pupil size, corneal aberrations, and lens alignment [20]. Optical bench tests for IOLs are becoming frequently used for the development of newer designs, for the preclinical verification of optical performance, and for the comparison of different types of IOLs. However, such a widely used bench test for IOLs has several

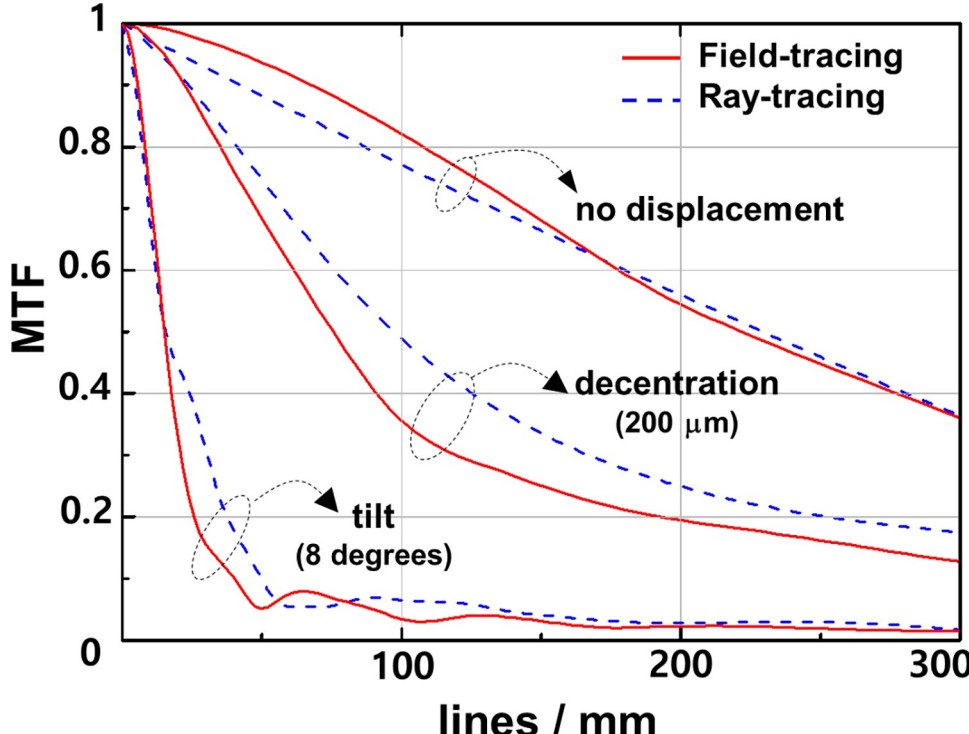

**Fig 4. The MTF values of the IOL by field tracing (solid red curve) and ray tracing (dashed blue curve) for the case of no misalignment, 200-μm decentration, and 8-degree tilt.** For the case of no misalignment, similar results were obtained for both of the tracing methods but not for the decentration or tilt.

disadvantages in being very time-consuming, having low flexibility for diverse optical environments, and relatively high cost for optical system construction.

Currently, optical design software has become available for modeling optical systems and for the design and optimization of optical elements such as IOLs [4]. It works by producing a mathematical description of the shapes, locations, and materials of all the optical elements used in a design. Optical modeling or simulation plays a central role in deriving new experimental knowledge for better understanding of the precise biological system and optical design of the eye [9]. Such software enables accurate and rapid virtual prototyping, allowing the performance of optical systems to be predicted and analyzed prior to fabrication. In general, this type of optical design software can be distinguished by ray-tracing simulation, which is based on geometrical optics and field-tracing simulation, which is based on the propagation of electromagnetic fields, that is wave optics [1,4].

Natural optical systems usually have non-smooth and irregular surfaces that need to be understood from a wave optical perspective. The dependence on the pupil or aperture is also important with any optic device independent of its optic design because, due to the nature of the light, even in a perfect system without aberrations, the effect of the light diffraction is unavoidable: the larger the pupil, the greater the amount of aberration effect, and the smaller the pupil, the greater the diffraction effect [21,22]. Ray tracing usually only allows for description of smooth surfaces and, in some cases, of periodical structures so as to propagate a specified number of optical rays from surface to surface for imaging. Similar results can be obtained for both ray- and filed-tracing methods as long as the diffractive effects are negligible; however, modern advanced optical systems usually have diffractive optical structures that cannot be ignored [4]. For example, multifocal IOLs based on arbitrary diffractive structures have to be analyzed by wave optical simulations, not by ray-tracing simulations, which cannot be properly modeled [4].

Fast Fourier transforms (FFTs) is a computationally efficient method of generating a Fourier transform to make the simulation of light field. Fraunhofer, Fresnel, Rayleigh-Sommerfeld, and Kirchoff solutions are well-known formulas for calculating diffraction [23]. But these formulas and FFTs have limitations because they are under the condition of the diffraction occurring in the no tilted screens or tilted aperture. Therefore, various formulas have been proposed to solve the error in the formulation and efficient computation of diffraction between two tilted planes [24]. For this reason, we think that the larger the tilted angle in Fig 3E, the greater the difference of the Strehl ratio between the ZEMAX simulation value and other values. Due to these problems, the field tracing method seems to be able to make more accurate predictions for the tilted IOL condition.

Wyrowskia and Kuhn [2] proposed that the field-tracing approach provides three fundamental advantages of practical concern: (1) field tracing enables unified optical modeling. Its concept allows the utilization of any modeling technique that is formulated for vectorial harmonic fields in different subdomains of the system; (2) the use of vectorial harmonic fields as a basis of field tracing permits great flexibility in source modeling; (3) in system modeling and design, the evaluation of any type of detector function is crucial. The use of vectorially formulated harmonic fields provides unrestricted access to all field parameters and therefore allows the introduction and evaluation of any type of detector.

In our study, we used a monochromatic laser as a light source. In a polychromatic light environment, the diffraction effect in the field-tracing simulation is decreased due to the blended effect among each wavelength of light. One might think that such a blended effect could make a comparison between ray-tracing and field-tracing simulation irrelevant. However, although the interference fringes from the target image would diminish, the diffraction effect among each wavelength of light would still exist and affect the image to some extent.

Eventually, for more rigorous reproduction of light propagation in an IOL bench test, field-tracing simulation is superior to ray-tracing simulation. In addition, according to the requirements of the ISO Standards 11979–2, the pupil size for optical evaluation of an intraocular lens should be 3 and 4.5 mm [25]. However, we had to choose the pupil sizes of 3 and 5 mm to compare with the previous study of Pieh et al. [17]. In general, the smaller the pupil size, the better overall optical quality results, so we chose the pupil size of 5 mm for the discriminating power of the results.

In conclusion, field-tracing optical bench simulation is a reliable method to evaluate IOL performance. It could provide retinal image quality close to real by accounting for the wave nature of light, interference and diffraction. Field-tracing optical bench simulation cannot replace the optical bench testing, but it enables the ophthalmologist to understand how the patient feels the diffractive optical system when having the misalignment of IOL by simulating the optical system. Furthermore, this can be useful for the patient to choose the IOL type before having real cataract surgery.

## Supporting information

**S1 Fig.**
(TIF)

**S1 Data.**
(XLSX)

## Author Contributions

**Formal analysis:** Seok Ho Song, Min Ho Kang.

**Investigation:** In Seok Song, Se Jin Oh.

**Methodology:** Seok Ho Song.

**Validation:** Min Ho Kang.

**Writing – original draft:** In Seok Song.

**Writing – review & editing:** Seok Ho Song, Hyeck-Soo Son, Min Ho Kang.

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
