## [Decision Letter · Decision Letter 0]

19 Aug 2021

PONE-D-21-11356

Optical bench simulation for intraocular lenses using field-tracing technology

PLOS ONE

Dear Dr. Kang,

Thank you for submitting your manuscript to PLOS ONE. After careful consideration, we feel that it has merit but does not fully meet PLOS ONE’s publication criteria as it currently stands. Therefore, we invite you to submit a revised version of the manuscript that addresses the points raised during the review process.

Please add the requested design data so others can try to reproduce your results. Please also answer the reviewer's comments about the amount of decentration that has been taken into account. Please also refer to the literature on the average amount of IOL decentration. 

It would help others to read about the differences/benefits of that Virtual Lab software over other optical simulation packages. 

We look forward to receiving your revised manuscript.

Kind regards,

Timo Eppig

Academic Editor

PLOS ONE

Journal Requirements:

Reviewers' comments:

Reviewer's Responses to Questions

**Comments to the Author**

1. Is the manuscript technically sound, and do the data support the conclusions?

Reviewer #1: Yes

2. Has the statistical analysis been performed appropriately and rigorously? 

Reviewer #1: I Don't Know

3. Have the authors made all data underlying the findings in their manuscript fully available?

Reviewer #1: No

4. Is the manuscript presented in an intelligible fashion and written in standard English?

Reviewer #1: Yes

5. Review Comments to the Author

Reviewer #1: 1. The design data of IOL in ZEMAX software is missing therefore it is not possible to simulate the study and check the results.

2. In figure 3e, There is so much difference in the strehl ratio value between ZEMAX simulation and other results. Where does this error come from?

3. in figure 4, Why do you compair 200µm decenteration with 8 degrees of tilt? 200µm in not maximum decenteration amount but 8 degrees of tilt is the maximum amount of tilt therefore I think it is not poosible to compaire the behaviour of tilt and decenteration.

6. PLOS authors have the option to publish the peer review history of their article (what does this mean?). If published, this will include your full peer review and any attached files.

Reviewer #1: No

---

## [Author Response · Author response to Decision Letter 0]

1 Oct 2021

#1 The design data of IOL in ZEMAX software is missing therefore it is not possible to simulate the study and check the results.

: Thank you for your important comments. As you have pointed out, we updated the design data as supporting information.

#2 In figure 3e, There is so much difference in the strehl ratio value between ZEMAX simulation and other results. Where does this error come from?

: Thank you for your valuable comment. Fast Fourier transforms (FFTs) is a computationally efficient method of generating a Fourier transform to make the simulation of light filed. Fraunhofer, Fresnel, Rayleigh-Sommerfeld, and Kirchoff solutions are well-known formulas for calculating diffraction. But these formulas and FFTs have limitations because they are under the condition of the diffraction occurring in the no tilted screens or tilted aperture. Therefore, various formulas have been proposed to solve the error in the formulation and efficient computation of diffraction between two tilted planes. We think that the calculation error occurring in the tilted IOL plane increases as the tilted angle increases. Due to these problems, the field tracing method seems to be able to make more accurate predictions for the tilted IOL condition. We added the above to the discussion.

#3 In figure 4, Why do you compare 200µm decenteration with 8 degrees of tilt? 200µm in not maximum decenteration amount but 8 degrees of tilt is the maximum amount of tilt therefore I think it is not poosible to compaire the behaviour of tilt and decenteration.

: Thank you for your valuable advice. To validate our study, we refered to the study of Pieh S, Fiala W, Malz A, and Stork W.(Reference number 17) They investigated the decentration at approximately 0.15 mm intervals from 0 mm to 0.75 mm, so we also investigated the decentration at 0.1 mm intervals from 0 mm to 0.8 mm. We compared the field-tracing and ray-tracing simulation in Figure 4 and presented the values of 200 µm decentration and 8 degrees tilt as representative examples of each. 

Additionally, we found and corrected small mistakes. The refractive index of acrylate is 1.47, but I wrote t incorrectly as 1.49.

 Again, thank you very much for your help with our manuscript.

Best regards,

Min Ho Kang, MD, PhD.

---

## [Decision Letter · Decision Letter 1]

27 Oct 2021

PONE-D-21-11356R1Optical bench simulation for intraocular lenses using field-tracing technologyPLOS ONE

Dear Dr. Kang,

Thank you for submitting your manuscript to PLOS ONE. After careful consideration, we feel that it has merit but does not fully meet PLOS ONE’s publication criteria as it currently stands. Therefore, we invite you to submit a revised version of the manuscript that addresses the points raised during the review process.

Please reply to the question of Reviewer 1 regarding the pupil size and include the information in the manuscript accordingly. 

We look forward to receiving your revised manuscript.

Kind regards,

Timo Eppig

Academic Editor

PLOS ONE

Journal Requirements:

Reviewers' comments:

Reviewer's Responses to Questions

**Comments to the Author**

1. If the authors have adequately addressed your comments raised in a previous round of review and you feel that this manuscript is now acceptable for publication, you may indicate that here to bypass the “Comments to the Author” section, enter your conflict of interest statement in the “Confidential to Editor” section, and submit your "Accept" recommendation.

Reviewer #1: (No Response)

2. Is the manuscript technically sound, and do the data support the conclusions?

Reviewer #1: Yes

3. Has the statistical analysis been performed appropriately and rigorously? 

Reviewer #1: Yes

4. Have the authors made all data underlying the findings in their manuscript fully available?

Reviewer #1: Yes

5. Is the manuscript presented in an intelligible fashion and written in standard English?

Reviewer #1: Yes

6. Review Comments to the Author

Reviewer #1: In the ''response to reviewers'' section, Supporting table 1, the diameter of the aperture is equal to 5 mm. According to the ISO standards the IOLs are tasted with 4.5 mm and 3 mm of aperture diameter.

Is there any specific reason for 5 mm in diameter for pupil size?

7. PLOS authors have the option to publish the peer review history of their article (what does this mean?). If published, this will include your full peer review and any attached files.

Reviewer #1: No

---

## [Author Response · Author response to Decision Letter 1]

8 Nov 2021

Reviewer #1: In the ''response to reviewers'' section, Supporting table 1, the diameter of the aperture is equal to 5 mm. According to the ISO standards the IOLs are tasted with 4.5 mm and 3 mm of aperture diameter.

Is there any specific reason for 5 mm in diameter for pupil size?

: Thank you for your valuable comment. We conducted the study comparing the results of Pieh S, et al. (Invest Ophthalmol Vis Sci 2009;50:1264-1270.) The study of Pieh S, et al. was conducted with pupil sizes of 3 and 5 mm. So we had to choose 3 and 5 mm pupil sizes instead of the ISO standard 3 and 4.5 mm as you pointed out. In general, the smaller the pupil size, the better overall optical quality results, so we chose the pupil size of 5 mm for the discriminating power of the results. We added the above to the discussion.

---

## [Editor Report · Decision Letter 2]

1 Dec 2021

Optical bench simulation for intraocular lenses using field-tracing technology

PONE-D-21-11356R2

Dear Dr. Kang,

We’re pleased to inform you that your manuscript has been judged scientifically suitable for publication and will be formally accepted for publication once it meets all outstanding technical requirements.

Kind regards,

Timo Eppig

Academic Editor

PLOS ONE

Additional Editor Comments (optional):

Please correct the following when submitting the final version for publication:

1) Please add the country to the address information brackets

2) manufacturer of Zemax is now ZEMAX LLC, Kirkland, WA, USA

3) manufacturer of VirtualLab is Wyrowski Photonics GmbH, Jena, Germany

4) page 10, end of page: ISO standard should be given without number delimiters: "of the standard ISO 11979-2"
---

## [Editor Report · Acceptance letter]

3 Dec 2021

PONE-D-21-11356R2 

Optical bench simulation for intraocular lenses using field-tracing technology 

Dear Dr. Kang:

I'm pleased to inform you that your manuscript has been deemed suitable for publication in PLOS ONE. Congratulations! Your manuscript is now with our production department. 

Kind regards, 

on behalf of

Dr. Timo Eppig 

Academic Editor

PLOS ONE